# Role of Mitochondrial Dynamics in Cocaine’s Neurotoxicity

**DOI:** 10.3390/ijms23105418

**Published:** 2022-05-12

**Authors:** Shuheng Wen, Toshihiko Aki, Takeshi Funakoshi, Kana Unuma, Koichi Uemura

**Affiliations:** Department of Forensic Medicine, Graduate School of Medical and Dental Sciences, Tokyo Medical and Dental University, 1-5-45, Yushima, Tokyo 113-8519, Japan; wenlegm@tmd.ac.jp (S.W.); funakoshi.legm@tmd.ac.jp (T.F.); unumlegm@tmd.ac.jp (K.U.); kuemura.legm@tmd.ac.jp (K.U.)

**Keywords:** cocaine, mitochondrial dynamics, mitochondrial fission, mitochondrial fusion, Drp1, endoplasmic reticulum

## Abstract

The dynamic balance of mitochondrial fission and fusion maintains mitochondrial homeostasis and optimal function. It is indispensable for cells such as neurons, which rely on the finely tuned mitochondria to carry out their normal physiological activities. The potent psychostimulant cocaine impairs mitochondria as one way it exerts its neurotoxicity, wherein the disturbances in mitochondrial dynamics have been suggested to play an essential role. In this review, we summarize the neurotoxicity of cocaine and the role of mitochondrial dynamics in cellular physiology. Subsequently, we introduce current findings that link disturbed neuronal mitochondrial dynamics with cocaine exposure. Finally, the possible role and potential therapeutic value of mitochondrial dynamics in cocaine neurotoxicity are discussed.

## 1. Introduction

Unity succeeds division, and division follows unity, such is the way with mitochondrial dynamics [1]. Though this famous double-membrane organelle is traditionally known for its bioenergetic function, mitochondria have profound influences on various physiological processes via morphological changes [2,3]. Mitochondrial dynamics participate in the regulation of mitochondrial quantity, quality, distribution, and metabolism [4]. When orchestrated in balance, this delicate mechanism maintains optimal mitochondrial output and cellular homeostasis, which allows the cell to resist intrinsic or extrinsic stress [5,6]. On the contrary, unbalanced mitochondrial dynamics leads to mitochondrial and cellular dysfunction, and is deeply involved in diverse pathological conditions [7]. It is particularly true for energy-intensive cells such as neurons, whose energy relies heavily on aerobic respiration [8]. Mitochondrial dynamics, by executing the bioenergetic function and participating in crucial cellular physiological processes, either independently or by interacting with other organelles, impact neurogenesis, neuronal plasticity, and functions including neurotransmission [9,10]. Disturbances in mitochondrial dynamics are a striking feature in the neurotoxicity of chemicals or drugs, including the potent psychostimulant cocaine [11]. Although indicating evidence suggested that altered neuronal mitochondrial dynamics may be closely associated with the appearance of clinical neurodegenerative symptoms [12], the causative role of defective mitochondrial dynamics in cocaine-induced neuronal pathogenesis remains to be illuminated. In this review article, we scrutinize the neurotoxicity of cocaine from the viewpoint of mitochondrial dynamics. We will describe the neurotoxicity of cocaine and the machinery of mitochondrial dynamics as well as its role in cellular physiology. We further introduce current findings that link disturbed mitochondrial dynamics with cocaine exposure on neurons. Finally, the possible role and potential therapeutic value of mitochondrial dynamics in the neurotoxicity of cocaine are discussed.

## 2. The Neurotoxicity of Cocaine

Cocaine (C_17_H_21_NO_4_) is a tropane alkaloid extracted from coca leaves. According to the latest report from the United Nations Office on Drugs and Crime [13], approximately 20,000,000 people used cocaine globally in 2019. This population represents a 22% increase in the last decade (Figure 1). The worldwide output of cocaine manufacture has also reached its highest level, with an especially sharp increase in Europe. Although the physiological effects of cocaine depend on the form, dose, administration route, and combination with other drugs [14], cocaine intake is associated with many mental or psychomotor disturbances [15,16] and detrimental brain consequences that can be life-threatening under some circumstances [17]. The detrimental effects of cocaine on the brain are comprehensively produced via multiple routes (Figure 2). Cocaine enters the central nervous system (CNS) either via direct crossing the BBB by passive diffusion and transport via proton-antiporter [18], or by destructing the blood–brain barrier (BBB) integrity through inducing acute hyperthermia and elevating the levels of brain serotonin [19,20]. The breakdown of the BBB leads to brain edema and neuronal damage in various brain regions [19], which contributes to the diverse brain consequences. The prolonged use of cocaine is also associated with accelerated brain aging, which may present as cognitive decline and brain atrophy [21,22]. A significant reduction in gray matter volume has been observed in chronic cocaine abusers [23]. By binding to monoamine transporters, cocaine blocks the reuptake of neurotransmitters (dopamine, serotonin, norepinephrine) and increases their release in synapses to carry out its strong stimulant effect [24,25]. The resultant constant and excessive synaptic concentration of dopamine leads to a dysregulated dopaminergic system and contributes to neurodegenerative processes [26]. Furthermore, the weak base property of cocaine enables its intracellular accumulation and direct effects on organelles such as mitochondria [27]. The neurotoxicity of cocaine results from the induction of subcellular stress in mitochondria and endoplasmic reticulum (ER), the following mitochondrial dysfunction, and the subsequent involvement of cell death pathways.

### 2.1. Overwhelmed Intracellular Oxidative Stress

Intracellular oxidative stress is a well-established feature of cocaine exposure. Mitochondria are not only the major organelle involved in the generation of reactive oxidative species (ROS) as a by-product of oxidative phosphorylation, but they are capable of scavenging ROS via their endogenous antioxidant enzymes including catalase, glutathione peroxide, and superoxide dismutase [5,28]. Overwhelming ROS generation that exceeds the endogenous antioxidant capacity contributes to oxidative stress and neuronal death [29]. Cocaine affects both the generation of stress and the antioxidant system. On one hand, during sustained excessive excitation of neurons, cocaine elevates ROS production within mitochondria by increasing calcium influx [30]. Its own degradation products, including norcocaine and norcocaine derivatives (nitroxide radical, N-hydroxy, nitrosonium, cocaine iminium, and formaldehyde), also have relatively higher oxidation potentials and result in stronger lipid peroxidation than cocaine induced itself [31]. Elevated oxidative stress has been well demonstrated in acute and chronic models of cocaine administration [32,33]. On the other hand, cocaine suppresses the expression and abolishes the activity of endogenous antioxidant enzymes and non-enzymatic antioxidants. The total antioxidant capacity was diminished as determined in blood samples of cocaine addicts [34]. Non-enzymatic antioxidants such as glutathione and vitamin E have also been found to be decreased after cocaine exposure [35,36]. The suppression of cocaine on antioxidant system has also been confirmed in in vivo and in vitro models involving single to chronic administration to withdrawal [35,37,38,39]. Aside from the direct effects, the aggregation of dopamine after cocaine exposure is considered another major route for intracellular oxidative stress formation. Dopamine results in the formation of ROS during its metabolism, either by monoamine oxidase-catalyzed intracellular oxidation or extracellular auto-oxidation [40]. Altogether, cocaine-induced oxidative stress leads to neuronal damage and is suggested to be connected to an increased risk of neurodegeneration. Aggregation of α-synuclein, an intrinsic disorder protein, results in various neurodegenerative changes related to oxidative stress and mitochondrial dysfunction; is the pathologic feature of neurodegenerative disorders known as synucleinopathies [41]. Increased expression of α-synuclein in the brain of cocaine abusers has been found and demonstrated to be a result of cocaine-induced dopamine accumulation and oxidative stress [42].

### 2.2. Mitochondrial Dysfunction

Mitochondrial dysfunction is considered to be a major event in cocaine neurotoxicity [43]. The canonical bioenergetic function of mitochondria relies on the steady operation of the electron transport chain (ETC) on the inner mitochondrial membrane. Electron transport during oxidative phosphorylation results in protons being pumped from the mitochondrial matrix to intermembrane spaces and the formation of an electrochemical gradient in the mitochondrial membrane [15]. Mitochondrial membrane potential is indispensable for bioenergetic functioning. Cocaine exposure directly impairs the enzyme activity of the ETC complexes via downregulating mitochondrial inner membrane and oxidative phosphorylation-related genes, as shown in different conditions including chronic administration and withdrawal [44,45]. Disruption of mitochondrial membrane potential and impaired respiration after cocaine administration has been illustrated in isolated mitochondria from rat brain, human and rat primary neurons [46,47,48].

Moreover, cocaine impacts the quality control mechanism of mitochondria. Mitochondrial biogenesis and mitophagy maintains the optimal quantity and quality of mitochondria by the coordination between the fresh formation of mitochondria and the degradation of unhealthy mitochondria via autophagosomes [49]. Under stress or injury conditions, both mitochondrial biogenesis and mitophagy are activated to preserve the normal mitochondrial function [50]. It is reasonable to be the same case with cocaine exposure. Brain region-specific increased expressions of mitochondrial biogenesis-related genes were found in a rat cocaine abuse model [51]. However, there is limited evidence to measure the exact performance of neuronal mitochondrial biogenesis after cocaine exposure. The machinery of mitochondrial biogenesis requires a well-organized collaboration between nuclear and mitochondrial genes [52], which can be difficult to accomplish under the damage of cocaine to mitochondrial genomes. Similarly, it remains unclear whether and how mitophagy participates against cocaine exposure on neurons. Supportive in vivo and in vitro findings in mouse microglial cells indicate an incomplete mitophagy in neurons under cocaine exposure with the deficient formation of autophagosomes, which results in the disordered mitochondrial quality control and leads to mitochondrial dysfunction [53].

Eventually, the overwhelmed oxidative stress and mitochondrial dysfunction lead to the activation of cell death pathways. By inducing oxidative stress and impairing oxidative phosphorylation, cocaine diminishes the mitochondrial membrane potential and causes the release of cytochrome c, resulting in neuronal apoptosis [35,46]. It has been found in neurons after cocaine exposure that acute or repeated administration triggers apoptosis via the upregulation of caspase-12 [54]. Cocaine-triggered apoptotic neuronal death has also been demonstrated in in vivo models in mouse, rat, and human [32,55,56]. Meanwhile, coordinating with mitochondria, ER stress induced by cocaine exposure via disrupting intracellular calcium homeostasis may also simultaneously induce apoptosis [57,58]. In addition, studies have suggested that autophagic cell death participates in the neurotoxicity of cocaine [59,60]. However, the role of cocaine-induced autophagy in neurotoxicity remains to be elucidated. Furthermore, mitochondria are also involved in other cell death pathways such as ferroptosis and necroptosis [61], yet whether these pathways play a role in cocaine-induced neurotoxicity needs further investigation.

As another key mechanism in mitochondrial quality control, mitochondrial dynamics have drawn increasing attention owing to the important and widely covered involvement in cellular physiology. With the progressing disclosure of the machinery of mitochondrial dynamics, growing evidence indicates the involvement and essential role of mitochondrial dynamics in the toxicity of cocaine, such as cardiotoxicity and neurotoxicity.

## 3. Machinery of Mitochondrial Dynamics

Mitochondria are bound by the outer and inner membranes. The inner membrane can be further divided into the inner boundary membrane, which runs parallel to the outer membrane and is distinguished by its intricate invaginated cristae where oxidative phosphorylation is conducted [62]. The changeable morphology of mitochondria, which varies from granules to unconnected or interconnected tubules, is governed by the continuous operating mechanism of fission and fusion, known as mitochondrial dynamics [8]. It is worth noting that the generalized concept of mitochondrial dynamics also encompasses mitochondrial trafficking, positioning, degradation, cristae biogenesis, and remodeling [8]. Herein, we will focus on the machinery of mitochondrial fission and fusion.

### 3.1. Mitochondrial Fission

Mitochondrial fission is an orchestration of multiple factors [63] (Figure 3). The major mediator is dynamin-related protein 1 (DRP1), a nucleotide guanosine triphosphate (GTP) hydrolyzing enzyme [64]. Most mitochondrial dynamics-related proteins, including DRP1 are encoded by nuclear genes [65]. At the beginning of fission, DRP1 is recruited from the cytosol to mitochondria by receptors in the outer membrane [66]. The cytosolic DRP1 exists in oligomeric states and organizes into higher-order spiral complexes after binding to its receptors [67]. Through its GTPase activity, the multimeric DRP1 winds around and constricts the mitochondria to scission [68].

The DRP1 receptors in the mitochondrial outer membrane consist mainly of mitochondrial fission factor (MFF) [69], mitochondrial protein of 51 kDa, (MiD51, also known as mitochondrial elongation factor 1) [70], mitochondrial protein of 49kDa (MiD49, also known as mitochondrial elongation factor 2) [71], and mitochondrial fission 1 protein (FIS1) [72]. It has been illustrated that these four receptors have distinct impacts on the fission process instead of functioning identically [73]. MFF, MiD51, and MiD49 are more prominent in the recruitment of DRP1 [74], and are largely associated with one another to form multimeric complexes [73]. On the other hand, FIS1 has been implied to interact with and suppress the fusion machinery, or mark fission sites, rather than collaborate with DRP1 in recruitment [75,76]. Although a MiDs level-dependent DRP1 regulation model has been proposed [77], the precise mechanism by which these four receptors coordinate in DRP1 recruitment still needs further clarification.

In order to modulate mitochondrial fission, DRP1 is regulated by multiple post-translational modifications to adapt its recruitment, polymerization, and activity to different cellular signals. These DRP1 modifications include phosphorylation or dephosphorylation, S-nitrosylation, SUMOylation [78], O-GlcNAcylation [79], and ubiquitination [80]. In particular, the different phosphorylation sites on DRP1, various kinases, and phosphates determine the activity of DRP1 based on cellular needs [81,82]. For instance, during processes such as entry to mitosis or tumor growth, the recruitment of DRP1 is strengthened by phosphorylation at Serine 616 [82,83]. Phosphorylation at Serine 637 suppresses DRP1 activity, whereas dephosphorylation at this site enhances it [84,85]. The ratio of phosphorylation at Serine 616 to that at Serine 637 has been suggested to reflect the exact activity of DRP1 under the added effects of diverse cues [86]. Apart from the classical machinery, complementary DRP1-independent mitochondrial fission mechanisms have been reported. In DRP1-deficient cells, FIS1 switches mitochondrial dynamics to favor fission by inducing the interaction between mitochondria and lysosomes or by inhibiting fusion mediators [75].

### 3.2. Mitochondrial Fusion

The machinery of mitochondrial fusion was well accepted over a decade ago. In the classical model, fusion starts with the tethering and fusing of the outer membranes from two apposing mitochondria by mitofusin1 and 2 (MFN1, 2) [87], and ends with the merging of the inner membranes from two sides mediated by optic atrophy 1 (OPA1) [88]. With progression in the understanding of proteins involved, the original impression of the fusion machinery has been challenged.

Similar to mitochondrial fission, three known major mediators of mitochondrial fusion belong to the dynamin superfamily with GTPase. The MFNs locate on the outer membrane and are associated with tethering and outer fusion [89,90]. To initiate mitochondrial fusion, two adjoining mitochondria tether together by binding the carboxyl-terminal of the opponent’s MFNs [87,91]. The mingled MFNs undergo conformational changes to create homogeneous or heterogeneous dimeric linkages that activate outer membrane fusion [92]. Although the two types of MFN have highly homologous structures [93], their functions are distinct [92]. MFN1 plays a central role in fusion machinery, while MFN2 seems to have more influence in other stages [94,95]. To adjust to a variety of cellular processes, the quantity and quality of MFNs are also modulated by post-translation modifications [96]. As of now, the mechanism for MFNs-mediated outer membrane fusion remains debated and awaits further elucidation.

OPA1, which is required for inner membrane fusion [97], anchors to the inner membrane and can be cleaved into a short form that aggregates in the mitochondrial matrix [98]. The exact mechanism by which the long and short forms of OPA1 mediate inner membrane fusion is controversial with differing interpretations. Some presume that the long form is adequate for promoting fusion while the short form may have other functions in inducing mitochondrial fission [99]. On the contrary, both isoforms have been indicated to possess independent fusion capacities [100]. Some argue that both forms of OPA1 are essential, and that an optimal balance is required for fusion depending on different conditions [101,102].

### 3.3. Mitochondrial Dynamics in Cellular Physiology

Although the relationship between mitochondrial fission and fusion literally appears to be biological opposite, it is more probable to be a mutual and cooperative partnership [103]. Certain mediators, such as DRP1 and MFNs, are even able to promote or compensate for a change in the other side to rebalance mitochondrial dynamics [104,105]. By working collectively, mitochondrial fission and fusion regulate bioenergetic metabolism [106], calcium homeostasis [107] and stress handling [108], and thereby contribute to cell proliferation [109], mitophagy, and cell death pathways [3,110].

Mitochondrial dynamics regulates bioenergetics of different paths. Fission is suggested to play an essential role in the assembly of ETC complexes [111]. Fusion mediators MFN2 and OPA1 also participate in ETC coupling, and OPA1 is further involved in oxidative phosphorylation by maintaining the cristae structure [112]. Consequently, disturbed mitochondrial dynamics affect the efficacy of mitochondrial respiration and the cellular energy supply [113]. In contrast, impaired mitochondrial respiration alters the dynamics and results in mitochondrial fragmentation [114].

Mitochondrial distribution is vital for the cellular energy supply because mitochondria should be trafficked and positioned within the cell so as to properly match the fluctuating energy demand in each region, especially in neurons with their extended axons and dendrites [115]. This is accomplished by the mitochondrial size regulation of dynamics, as well as the direct involvement of dynamics mediators in trafficking [116]. In neurons, mitochondrial distribution depends on the interactions between anchoring proteins in the mitochondrial outer membrane, the motor-adaptor proteins, and the motor proteins on microtubules, for example, the Miro–Milton-dynein interaction [117]. Owing to their tethering function, MFNs are essential effectors in interaction-mediated mitochondrial trafficking [118]. Mfn mutation-induced neurodegenerative disease has been attributed to the impairment in MFN2-related mitochondrial trafficking [119].

Furthermore, balanced mitochondrial dynamics are essential for the inheritance and integrity of mtDNA [120]. Despite most mitochondrial proteins being encoded by the nuclear genome, the mitochondrial DNA (mtDNA) genome encodes 13 proteins requisite for oxidative phosphorylation [8]. The mtDNAs are vulnerable and prone to mutations because of their relatively short half-life, sustained exposure to oxidative stress, and lack of protection or repair mechanisms [121,122]. Aggregating with aging, higher-level mtDNA mutations in the brain lead to impaired oxidative metabolism and mitochondrial dysfunction [123]. Mitochondrial fission enables segregation of mutant or damaged mtDNA for further degradation [124]. In addition, mtDNA determines the sites for mitochondrial fission so as to separate the replicated mitochondrial genome [125,126]. Fusion, on the other hand, provides toleration of mtDNA mutations by exchanging or diluting the detrimental effects among mitochondria [127]. Accordingly, disrupted mitochondrial fusion results in an elevated mutation rate and depletion of mtDNA [128]. In addition to mtDNA nucleoids, mitochondrial fusion allows the exchange of proteins, ETC complexes, and lipid membrane, so as to limit defective mitochondrial components to a tolerable level and maintain the homogenization of mitochondria [129,130]. This has been proposed as one of the foremost functions of mitochondrial fusion [131].

Moreover, mediators of mitochondrial dynamics are associated with mitophagy. When a mitochondrion is depolarized, PTEN-induced kinase 1 (PINK1) may induce the phosphorylation of MFN2 and the translocation of Parkin [96,132]. It has been suggested that, in the brain, MFN2 acts as a fusion component while a mitophagy effector or receptor of Parkin after phosphorylation induced by PINK1 [12]. The recruited Parkin ubiquitinates and induces the degeneration of MFN2 to inhibit further fusion of depolarized mitochondria and initiate mitophagy [133]. Meanwhile, DRP1-dependent fission separates the impaired mitochondria to allow further engulfment of autophagosomes to advance mitophagy [49].

Mitochondrial dynamics also participate in the regulation of apoptosis [134]. DRP1 has been proposed as a precursor of apoptosis, especially in neurodegenerative pathologies [124]. Overwhelming mitochondrial impairments, such as high-level DNA damage, outraged oxidative stress, or depolarization, trigger DRP1-mediated mitochondrial fragmentation and activate apoptosis [3]. Once apoptosis is initiated, FIS1 mediates the translocation of the cytosolic apoptosis regulators to the mitochondrial outer membrane [135]. The recruited apoptosis regulators colocalize with DRP1 or MFN2 and undergo oligomerization induced by DRP1 or MiD51 [136,137]. Simultaneously, OPA1 mediates the remodeling of mitochondrial cristae and the release of cytochrome c to the cytosol [138]. Collectively, these result in mitochondrial fragmentation and further apoptosis. Although fission is a basic event in apoptosis [135], the precise role of DRP1 in this intricate process remains to be elucidated. Apart from apoptosis, mitochondrial dynamics have been implied to be involved in other cell death pathways such as necroptosis and ferroptosis [61]. Altogether, the balance between mitochondrial fission and fusion cooperatively modulates diverse cellular physiological processes.

### 3.4. Mitochondrial Dynamics and Endoplasmic Reticulum

Mitochondria also interact with other organelles concomitantly instead of participating in cellular physiology independently. In particular, the crosstalk between mitochondria and ER has drawn increasing attention [139,140]. ER functions in protein folding and post-translational modifications, as well as in the maintenance of intracellular Ca^2+^ homeostasis [141]. The membranes of ER are capable of associating with the mitochondrial outer membrane by protein tethering [142] through what are known as mitochondria–ER contact (MERC) sites [143]. Approximately 5~20% of the mitochondrial outer membrane is connected to ER [144].

MERC is important for the exchange of content between the two organelles [145]. Intracellular calcium has a profound influence on a variety of cellular processes, and it is mainly centralized in mitochondria and ER. MERC provides a bidirectional lane for calcium influx, and enables mitochondria and ER to efficiently maintain cellular calcium homeostasis [146]. This calcium exchange is not only significant for mitochondrial bioenergetic function [147], but implicate the apoptotic pathway under pathological conditions [2]. The content exchange also includes lipids, which are associated with mitochondrial structural stability and functions [148]. Moreover, MERC may involve in protein import, mitochondrial distribution, and mitophagy [139]. Consequently, MERC can play a protective role under adverse conditions. When stress accumulates in ER, MERC sites are found to increase [149].

Intriguingly, many mitochondrial dynamics-related mediators are connected to MERC, in which MFN2 is suggested to play a key role. MFN2 is expressed in both the mitochondrial outer membrane and ER, and it constructs a bridge during their association by forming complexes with MFNs located on the mitochondrial outer membrane [150]. MFN2 is capable of promoting calcium uptake and further affect ER functions through MERC [151]. MFN2 ablation results in unstable MERC and deficits in mitochondrial calcium influx [95]. It should be noted that the topological determination of MFNs is still in progress, and the requisite role of MFNs in MERC has also been questioned [152,153]. Moreover, FIS1 has been implied to be involved in the regulation of this association [154].

Moreover, MERC participates directly in mitochondrial dynamics. This structure tethering seems to determine the position of mitochondrial fission. In a majority of the fission initiated stages, ER tubules have been found to wrap around the mitochondrion and constrict to reduce the diameter of the mitochondrion before the recruitment of DRP1 [155], after which fission mediators, including DRP1, MFF, MiD51, and MiD49, may colocalize at these sites with other effectors so as to advance the division process [156,157]. Meanwhile, MERC facilitates the mtDNA distribution during mitochondrial fission by coupling the replicated mtDNA nucleoids [126]. Disturbances in ER function are commonly observed in neurodegenerative diseases [158]. The possible implication of MERC in the pathogenesis of these neurodegenerative disorders has been proposed, in which calcium communication and oxidative stress are considered to play key roles [159,160]. The close interaction and coordination between mitochondria and ER in neurons should provide another appealing topic for further research concerning the neurodegenerative diseases and neurotoxicity of drugs.

To summarize, fine-tuned mitochondrial dynamics are closely connected with the appropriate operation of mitochondria and cellular fitness. Neurons rely heavily on mitochondrial oxidative phosphorylation for energy [53]. They have unique morphologies with extensive differentially extended arborizations and constantly remodeling synapses that demand sufficient energy supply and delicate calcium regulation [54,55]. All these characteristics make orchestrated mitochondrial dynamics pivotal for the normal functions of neurons. The balance of mitochondrial dynamics as a compensating mechanism is conversely governed by a variety of physiological signals and is sensitive to pathological or detrimental stimuli [161,162]. Disruption of this equilibrium leads to accumulated impaired mitochondrial fragments, and it is related to neurodegenerative pathologies [49,117]. Therefore, it is fascinating to explore the role of mitochondrial dynamics in the neurotoxicity of cocaine.

## 4. Mitochondrial Dynamics and the Neurotoxicity of Cocaine

Mitochondrial fragmentation resulting from disturbances of mitochondrial dynamics including enhanced DRP1-dependent mitochondrial fission and weakened mitochondrial fusion is a well-established feature that leads to mitochondrial dysfunction and neurodegeneration in common neurodegenerative diseases [163,164]. Indicative evidence has also been found in our former study on the cardiotoxicity of cocaine [165]. Before the occurrence of any structural changes in the myocardium, cocaine exposure to cardiomyocytes altered the mitochondrial dynamics with a similarly increased fission and decreased fusion as is found in neurodegenerative diseases. However, at this time, studies concentrated on the involvement of mitochondrial dynamics in the neurotoxicity of cocaine can be counted on the fingers of one hand. Sadakierska-Chudy et al. established a cocaine self-administration model (average of 100 mg/kg of cocaine intake within 3 days) in rats following short-term abstinence (3 days) to study the potential role of mitochondrial dynamics in the reward circuit and drug-seeking behaviors [166]. Using microarray and quantitative real-time PCR assays, they determined a significantly increased copy number of mtDNA and elevated expressions of nuclear-encoded mitochondrial genes associated with oxidative phosphorylation and mitochondrial dynamics. Increased fission was identified in the hippocampus whereas increased fusion was found in the prefrontal cortex (e.g., Mfn1, Opa1). Correspondingly, upregulated ER stress-induced genes were also revealed. Although the authors considered the increased mtDNA copy number as a compensative role against intracellular stress, it should be noted that the meaning of these findings can be indefinite, since neither change in relative protein levels nor the mitochondrial function were determined. Still, this study suggested the involvement of mitochondrial dynamics in early stage of the neurotoxicity of cocaine and provided indicative findings on possible alterations in the mitochondrial dynamics of different brain regions after short-term self-administered cocaine exposure.

In another study, Chandra R et al. associated the transcription factor early growth response 3 (Egr3) in nucleus accumbens (NAc) with cocaine reward and locomotor responses by applying an in vivo cocaine abuse model (20 mg/kg/day, intraperitoneal administration for 7 days) [167]. Erg3 is induced by acute cocaine exposure and has been found pivotal in cocaine-related signaling pathways. They found the overexpression of Egr3 in D1 type medium spiny neurons elevates the cocaine reward and locomotor responses while its overexpression in D2 type medium spiny neurons inhibits these responses. In the following research, they demonstrated the requisite role of Erg3 in facilitating mitochondrial fission in NAc after cocaine exposure utilizing intraperitoneal administration (20 mg/kg/day for 7 days), self-administration (1 mg/kg/infusion for 10 days) models, and postmortem brain samples of cocaine addicts. They found the binding of Erg3 to Drp1 was increased in NAc during cocaine exposure, whereas the Erg3 knockdown attenuated cocaine-induced mitochondrial fission [168]. These studies throw light on the potential involvement of mitochondrial dynamics in the behavioral modulation of cocaine abuse.

Subsequently, Chandra R et al. focused on the expressions of dynamics-related mediators in the NAc in mice and rats after cocaine exposure, and in the brains of cocaine addicts [169]. They found increased levels of NAc Drp1 mRNA in mice after 7 days of cocaine administration (intraperitoneal injection, 20 mg/kg) as well as in self-administered cocaine rats (1 mg/kg/infusion for 10 days). A similar increased expression of NAc Drp1 mRNA was also observed in the brains of cocaine addicts. Similarly, Drp1 has also been shown to be differentially regulated by cocaine in D1 type and D2 type medium spiny neurons, where Drp1 expression is upregulated in D1 type yet downregulated in D2 type medium spiny neurons upon cocaine exposure. Further, they found increased phosphorylation of DRP1 at serine 616 and a corresponding decrease in mitochondrial size, specifically in dopamine receptor 1-containing medium spiny neurons after cocaine self-administration. Moreover, by applying DRP1 inhibitor Mdivi-1 and Drp1 knockdown, they linked DRP1-dependent mitochondrial fission in dopamine receptor 1-containing medium spiny neurons to the development of cocaine-addictive behavior. Interestingly, there seems to be a threshold for DRP1-dependent mitochondrial fission in dopamine receptor 1-containing medium spiny neurons during cocaine administration. The enhancement of cocaine-seeking induced by promoted DRP1-dependent mitochondrial fission appeared to hit a limit at the end of the self-administration session, while the abstinent period restored the enhancement of DRP1-dependent mitochondrial fission on cocaine-seeking. This could reflect that the elevated mitochondrial fission during cocaine exposure has fallen back to the basal level. Due to its indispensable role in mitochondrial fission, DRP1 is important in neuronal maturation, synaptic neurotransmission, and neuronal plasticity [10,170]. Posttranslational modifications in DRP1 are related to neuronal injury under stress conditions [171]. Activated DRP1 causes an elevation in mitochondrial fission in affected neurons, leading not only to mitochondrial fragmentation, but also to impaired mitochondrial axonal distribution and synaptic degeneration resulting in neuronal damage [171,172]. This study provided solid evidence about the behavioral function of DRP1-dependent mitochondrial fission and its therapeutic value in cocaine abuse.

In another study, Funakoshi T. et, al. provided supporting evidence for the involvement of altered mitochondrial dynamics in the neurotoxicity of cocaine. They examined the effect of cocaine on mitochondrial dynamics in an in vitro model using Neuro2a neuroblastoma cells (600 mM, 2~3 times per week, 3 weeks) [11]. They found increased phosphorylation of DRP1 at serine 616 and the expression of Fis1 mRNA with a consequential decrease in mitochondrial size with the help of transmission electron microscopy. In addition, they also reported a decrease in mitochondrial membrane potential and activated autophagy as well as an elevation of Parkin after chronic cocaine exposure. Further clarification of the relationship between increased mitochondrial fission and mitochondrial dysfunction, as well as the potential involvement of autophagy after neuronal cocaine exposure should be worth pursuing. The cocaine’s effects on neuronal mitochondrial dynamics introduced in this section have been summarized in Table 1.

It has been suggested that mitochondrial fission might play a protective role against the detrimental effects of cocaine. Specific alterations of mitochondrial dynamics appear to be dependent on different brain regions or cell types, which indicates a potential role of mitochondrial dynamics in addiction and to a wider stage.

## 5. Conclusions

Insights into the causality and role of mitochondrial dynamics in neuronal degeneration should be beneficial for understanding pathogenesis and developing therapeutic strategies. Considering the implied role of DRP1-mediated fission and mitochondrial fragmentation in cocaine addiction [166], it would be enchanting to determine the participation of other key mitochondrial dynamics-related mediators and the potential regulating mechanisms in the neurotoxicity of cocaine under different experimental conditions.

Optimal mitochondrial morphologies and ultrastructure are correlated with multiple conditions including bioenergetic demands, cellular milieu, or occupied cell types [49,173]. Correspondingly, the profiles of balanced mitochondrial dynamics and mitochondrial morphology can differ greatly among brain regions [174]. Thus, the diagnosis of impaired dynamics should be based on comprehensive determinations including brain regions, dynamics profiles (including in-depth indexes such as the ratio of differently phosphorylated DRP1, or the ratio of short-form to total OPA1) as well as measurements of mitochondrial size. Deficits in mitochondrial dynamics appear to be systematic, as found in peripheral blood lymphocytes in neurodegenerative disorder patients [175,176]. It is reasonable to presume a similar consequence in cocaine addicts. Furthermore, since Midiv-1 has proved effective in attenuating cocaine addiction and its neurotoxicity in an in vivo model [169], it would be compelling to learn whether other agonists and antagonists of mitochondrial dynamics might have similar therapeutic effects in cocaine abuse models. However, it should be noted that both fission and fusion are essential elements for the normal functioning of neuronal mitochondria [124], so that a complete inhibition of either might have unexpected detrimental effects. As a result, no matter whether using drugs or gene strategies, therapeutic attempts should determine a suitable dose and target to achieve minute adjustments in mitochondrial dynamics and restore the balance between fission and fusion. In this review article, we have limited our focus to the involvement of mitochondrial dynamics in common cellular physiological processes and cocaine toxicity. It should be noted that mutations in dynamics-related genes also contribute to impaired neuronal development and neurodegenerative diseases, such as autosomal dominant optic atrophy and Charcot-Marie-Tooth neuropathy [177].

In conclusion, we introduced the neurotoxicity of cocaine and the machinery of mitochondrial dynamics. We further presented current findings that associate mitochondrial dynamics with neuronal exposure to cocaine. Balanced mitochondrial dynamics is indispensable for neuronal health. With the disclosure of cell signaling affecting dynamics, efforts to obtain a deeper understanding of mitochondrial dynamics are progressing. We hope this article provides beneficial information and outlooks for further studies on the role of mitochondrial dynamics in the neurotoxicity of cocaine.

## Figures and Tables

**Figure 1 ijms-23-05418-f001:**
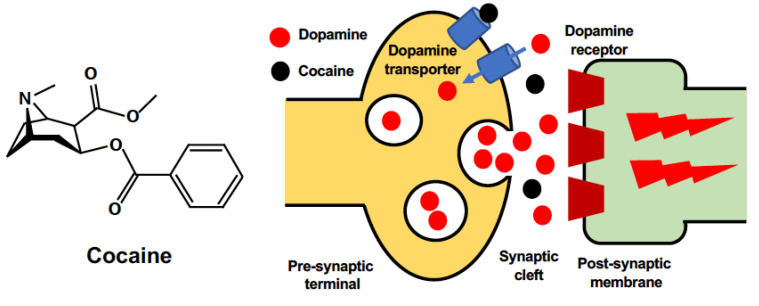
Chemical structure of cocaine and its action on dopaminergic neurons. Cocaine antagonizes dopamine reuptake from the synaptic cleft through dopamine transporters. This leads to an increase in the levels of dopamine in the cleft, boosting the effects of dopamine in the central nervous system.

**Figure 2 ijms-23-05418-f002:**
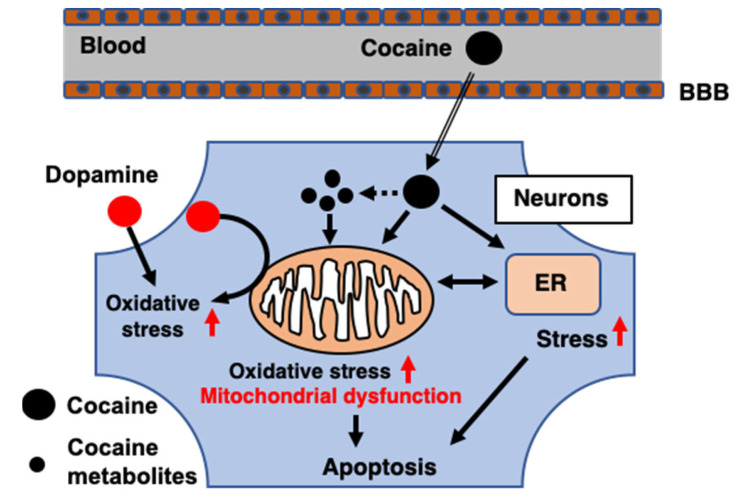
Effects of cocaine on neuronal cells. Cocaine enters the central nervous system by crossing and/or disrupting the blood–brain barrier (BBB). After entering neuronal cells, cocaine and its metabolites induce subcellular stress within mitochondria and the endoplasmic reticulum (ER). The aggregation of dopamine induced by cocaine also results in oxidative stress during the degradation of dopamine. Overwhelmed stress and direct damage from cocaine lead to mitochondrial dysfunction which results in cell death.

**Figure 3 ijms-23-05418-f003:**
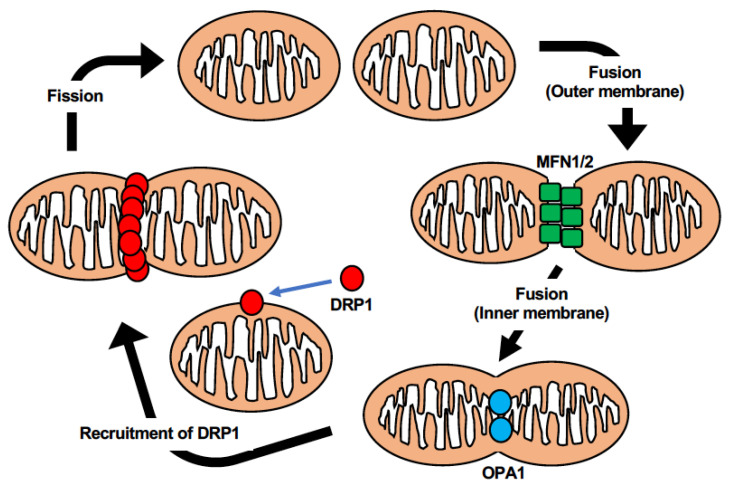
General aspects of mitochondrial fission and fusion mediated by a panel of regulating molecules. DRP1, which ordinarily resides in cytosol, is recruited to mitochondrial outer membranes upon activation through, e.g., phosphorylation at Ser-616. DRP1 assembles into a ring-like structure and act as scissors to promote mitochondrial division. Mitochondrial fusion consists of two consecutive processes: outer and inner membrane fusions. The former process is mediated by MFN1/2, while the latter is mainly executed through OPA1.

**Table 1 ijms-23-05418-t001:** Current findings of how cocaine affects neuronal mitochondrial dynamics.

Mitochondrial Dynamics	Altered Markers	Tested Brain Region/Cell Type	Model Details
Fission related	*Mtfr1* (↑)	Prefrontal cortex	Rat, self-administration (average 100 mg/kg of cocaine intake, 3 days) and abstinence (3 days) [166]
	*Opa3* (↑)	Prefrontal cortex	
	*Drp1* (↑)	Nucleus accumbens	Mice, repeated cocaine (7 days, 20 mg/kg, ip) [169]
		Nucleus accumbens	Rat, self-administration (1 mg/kg/infusion, 10 days) [169]
		Nucleus accumbens	Human samples [169]
		D1 type medium spiny neurons	Mice, repeated cocaine (7 days, 20 mg/kg, ip) [169]
	*Drp1* (↓)	D2 type medium spiny neurons	
	pDRP1 at ser616(↑)	D1 type medium spiny neurons	
		Neuro2a neuroblastoma cells	600 mM, 2~3 times per week, 3 weeks [11]
	pDRP1 at ser616 (↓)	D2 type medium spiny neurons	Mice, repeated cocaine (7 days, 20 mg/kg, ip) [169]
	Mitochondrial size (↓)	D1 type medium spiny neurons	Rat, self-administration (1 mg/kg/infusion, 10 days) [169]
		Neuro2a neuroblastoma cells	600 mM, 2~3 times per week, 3 weeks [11]
Fusion related	*Mfn1* (↑)	prefrontal cortex	Rat, self-administration (average 100 mg/kg of cocaine intake) and abstinence (3 days) [166]
	*Opa1* (↑)	prefrontal cortex	

*Mtfr1*, mitochondrial fission regulator 1, *Opa3*, optic atrophy 3, *Drp1*, dynamin-related protein 1, pDRP1 at ser616, phosphorylation of DRP1 at serine 616, *Mfn1*, mitofusion 1, *Opa1,* optic atrophy 1, ip, intraperitoneal injection.

## Data Availability

Not applicable.

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
