# Peer review of "Role of Mitochondrial Dynamics in Cocaine’s Neurotoxicity"

_ijms, 2022, doi:10.3390/ijms23105418_

Round 1

Reviewer 1 Report

Peer Review “Role of Mitochondrial Dynamics in Cocaine’s Neurotoxicity”

Line 21-22. Love the beginning of the article

Line 87.  …ROS production by increasing calcium influx….  As I read I was curious which organelle, I think its safe to assume its mitochondria but maybe clarify that this statement is referring to mitochondria

Line 95-96. Maybe add something about what a-synuclein does and its role in neurodegeneration? I feel like this sentence doesn’t follow the previous one well.

Line 122. Try  “Brain region-specific increased expressions of…” delete ‘The’

Line 123. I think it should say in “a” rat cocaine abuse model?

Line 126. Fulfill is an odd word choice here, maybe accomplish?

Line127. Unsure what is being stated, maybe measure or quantify instead of present?

Line 128. Are these findings still in the rat model?

Line 125-150. In general this was the hardest to follow section, the rest of the manuscript is well written but this is a bit confusing and slightly choppy. Its ok but if the authors can clean it up it would be helpful.

Line 366. Replace “with the findings” with as is found

Line 380. Starting the sentence with “Even though, this study” is confusing. I think it would read better if you used “Still, this study..” Or something similar.

Line 390. I had a hard time following the goal/results of this study, maybe explain how they know Erg3 facilitates mitochondrial fission?

Line 406. I think this could use more clarification, what do you mean by threshold?

Line 430. Should be  dependent instead of depending and indicates instead of indicated

Line 442. I know I have read another paper from the Fisher Lab at LaTrobe University that supports this.  I can’t find the citation right now, but the review by Annesely and Fisher, IJMS 2021 Lymphoblastoid cell lines as models to study mitochondrial function in neurological disorders would likely be a great additional citation.

Line 444-446. I think MERC’s possible role in pathogenesis of neurodegenerative disorders should be discussed a little in the last paragraph of section 3.4 – I didn’t really understand why section 3.4 was included until I saw this in the conclusion.

Line 462. Change to “unexpected detrimental effects”

Reviewer 2 Report

The manuscript of “Role of Mitochondrial Dynamics in Cocaine’s Neurotoxicity” by Wen S. and co-authors aims to summarize the neurotoxicity of cocaine and the role of mitochondrial dynamics in this process. The manuscript makes a significant contribution to the systematization of knowledge about mitochondrial dynamics. The authors have cited a large number of research articles, a significant portion of which have been published over the last five years. However, the manuscript contains too many sections with general information about mitochondrial dynamics (Sections 3. Machinery of Mitochondrial Dynamics 3.1 Mitochondrial fission 3.2 Mitochondrial fusion 3.3 Mitochondrial dynamics in cellular physiology 3.4 Mitochondrial dynamics and endoplasmic reticulum) without focusing on the research topic. The manuscript may be accepted for publication after major revision.

Major concerns:

  1. In the abstract section, the authors mentioned that they introduce “current findings that link disturbed neuronal mitochondrial dynamics with cocaine exposure”. Unfortunately, only a small part of the manuscript is devoted to this issue (Section 4). Most of the text is devoted to general ideas about mitochondrial dynamics in cellular physiology. In this regard, the author should try to reduce the well-known facts about mitochondrial dynamics and describe in more detail the mechanisms of cocaine-induced mitochondrial injury and neurotoxicity, as well as discuss the role of impaired mitochondrial dynamics in these processes. The effects of cocaine on mitochondria and /or mitochondrial dynamics should be summarized in a table (with references) or an illustrative diagram.
  2. The section “2.1 Overwhelmed intracellular oxidative stress” describes only mitochondrial antioxidant enzymes. It is necessary either to specify the title, or to describe all the mechanisms of oxidative stress in the cell when exposed to cocaine. It is also necessary to add information about non-enzymatic antioxidants in mitochondria (and, perhaps, their changes after cocaine exposure). The authors mentioned that “cocaine elevates ROS production by increasing calcium influx, and its own degradation products have relatively higher oxidation potentials [30]” (Lines 87-88). These findings and specific sources of mitochondrial ROS could be described in more detail.
  3. Lines 429-431: The authors concluded that “specific alterations of mitochondrial dynamics appear to be depending on different brain regions or cell types”. However, these issues were poorly discussed in the text.
  4. The Concluding Remarks section is quite large and looks like a continuation of the main text. It discusses different mechanisms, types of cells, and neurodegenerative diseases, which does not allow the reader to draw a conclusion about the potential therapeutic value of mitochondrial dynamics in cocaine neurotoxicity. This section needs to be more specific.

Round 2

Reviewer 2 Report

The manuscript “Role of Mitochondrial Dynamics in Cocaine’s Neurotoxicity” by Shuheng Wen and co-authors has been significantly improved. The authors have addressed all my concerns. The revised version of the manuscript may be accepted for publication.